# PPP3CB Inhibits Migration of G401 Cells via Regulating Epithelial-to-Mesenchymal Transition and Promotes G401 Cells Growth

**DOI:** 10.3390/ijms20020275

**Published:** 2019-01-11

**Authors:** Lei Chen, Qingling He, Yamin Liu, Yafei Wu, Dongsheng Ni, Jianing Liu, Yanxia Hu, Yuping Gu, Yajun Xie, Qin Zhou, Qianyin Li

**Affiliations:** The Ministry of Education Key Laboratory of Clinical Diagnostics, School of Laboratory Medicine, Chongqing Medical University, Chongqing 400016, China; chenlei@stu.cqmu.edu.cn (L.C.); hqltsang@foxmail.com (Q.H.); liuyamin2013@126.com (Y.L.); wuyafei2011@gmail.com (Y.W.); dongshengni@outlook.com (D.N.); keithljn@sina.com (J.L.); hyx_zuichu@outlook.com (Y.H.); guyupinglittle@outlook.com (Y.G.); xieyajun2007@163.com (Y.X.); zhouqin@cqmu.edu.cn (Q.Z.)

**Keywords:** epithelial–mesenchymal transition, PPP3CB, metastasis, proliferation

## Abstract

PPP3CB belongs to the phosphoprotein phosphatases (PPPs) group. Although the majority of the PPP family play important roles in the epithelial-to-mesenchymal transition (EMT) of tumor cells, little is known about the function of PPP3CB in the EMT process. Here, we found PPP3CB had high expression in kidney mesenchymal-like cells compared with kidney epithelial-like cells. Knock-down of PPP3CB downregulated epithelial marker E-cadherin and upregulated mesenchymal marker Vimentin, promoting the transition of cell states from epithelial to mesenchymal and reorganizing the actin cytoskeleton which contributed to cell migration. Conversely, overexpression of PPP3CB reversed EMT and inhibited migration of tumor cells. Besides, in vitro and in vivo experiments indicated that the loss of PPP3CB suppressed the tumor growth. However, the deletion of the phosphatase domain of PPP3CB showed no effect on the expression of E-cadherin, migration, and G401 cell proliferation. Together, we demonstrate that PPP3CB inhibits G401 cell migration through regulating EMT and promotes cell proliferation, which are both associated with the phosphatase activity of PPP3CB.

## 1. Introduction

Epithelial–mesenchymal transition (EMT) plays a critical role in promoting tumor invasion and metastasis [1]. During the EMT process, the epithelial tumor cells lose cell–cell adhesion and cell polarity, change cellular morphology and reorganize the actin cytoskeleton, and gain the capabilities of invasion and migration [2]. Multiple signaling pathways, such as TGF-β, MAPK, WNT, NOTCH, key transcription factors (Twist1, Snail1, Zeb1), and miRNAs, form a complex signaling network to modulate the process of EMT [3,4,5]. Vital hallmarks change in the process of EMT, such as downregulation of epithelial markers (E-cadherin) and the upregulation of mesenchymal marker (Vimentin) [6]. The PPP family has seven members, including Protein Phosphatases1 (PP1), 2A (PP2A), and 2B (PP2B) [7]. These members are involved in EMT through regulating the expression of EMT markers and modulating a broad range of signaling pathways. For example, loss of PPP2CA reduces the expression of E-cadherin and elevates the expression of Vimentin in prostate cancer cells, leading to migration via activating β-catenin and NF-κB [8]. PP2B-NFAT signaling activates Twist1 and Snail1 and redistributes E-cadherin to induce EMT in embryonic stem cells through increasing src expression [9]. Activation of PPP3CA is reported to induce EMT and produce cancer stem cells, thus promoting the metastatic processes of breast cancer [10]. Therefore, these reports indicate that the phosphoprotein phosphatase (PPP) family plays vital roles in EMT. However, the roles of individual PPP family members in EMT remain largely unknown.

PPP3CB, the key member of the PPP family, comprises four domains: a catalytic domain, a calcineurin B-binding segment, a calmodulin-binding segment, and an autoinhibitory segment [11,12]. PPP3CB is ubiquitously expressed, particularly in the brain and kidney, and is associated with the pathogenesis of renal diseases [12,13,14]. For example, PPP3CB causes cyclosporineA (CsA)-induced secretion MMP-9, which contributes to renal fibrosis [13]. The upregulation of PPP3CB promotes the apoptosis of podocytes [15]. Our quantitative proteomics analysis results show that PPP3CB is highly expressed in kidney mesenchymal-like cells compared with kidney epithelial-like cells. This suggests that PPP3CB may be a potential regulator of EMT in the kidney. However, the function of PPP3CB in the EMT of kidney tumors has not been explored.

Herein, we demonstrate that PPP3CB overexpression inhibits EMT and migration of G401 cells and promotes tumor proliferation. Depletion of PPP3CB leads to acquisition of a mesenchymal state, thus enhancing cell migration. Knocking down of PPP3CB significantly represses G401 cells’ growth in vivo and in vitro. Additionally, the deletion of PPP3CB 1–401 has no impact on EMT and cell proliferation. Altogether, PPP3CB suppresses EMT of G401 cells and enhances cell proliferation.

## 2. Results

### 2.1. PPP3CB is an Important Regulator in Kidney

To screen unknown regulators of EMT via mass spectroscopy from kidney epithelial-like cells (mK4) and kidney mesenchymal-like cells (mK3), differentially expressed proteins were selected by volcano plot filtering (*p*-value ≤ 0.05) as shown in Figure A1A. We found some candidates which might regulate EMT from the analysis results. Among these candidates, PPP3CB served as a potent regulator of EMT (Figure A1A). To determine the analysis result, we tested the expression of PPP3CB in mK3 and mK4 cells. Results of Quantitative polymerase chain reaction (QPCR) demonstrated that PPP3CB had high expression in mK3 cells compared with mK4 cells (Figure 1A). In accordance with the mRNA level, the PPP3CB protein was also highly expressed in mK3 cells compared to mK4 cells (Figure 1B). These results indicate that PPP3CB is potentially involved in EMT. In a recent study, PPP3CB was found in various tissues, particularly in the brain and kidney [13]. We examined the distribution of PPP3CB in the different tissues of mouse. As expected, PPP3CB was highly expressed in brain, heart, and kidney (Figure 1C,D). To further analyze the distribution of PPP3CB in human tissues, we used bioinformatic analysis. Analysis of The Human Protein Atlas revealed high PPP3CB expression in the cerebral cortex and lung and medium expression of PPP3CB in kidney (Figure A1B). Together, all of results indicate that PPP3CB may play an important role in kidney.

### 2.2. PPP3CB Suppresses EMT of G401 Cells

PPP3CB is a member of the PPP family. The majority of the PPP family regulate the process of EMT, but the role of PPP3CB in EMT remains largely unclear. As mentioned above, PPP3CB plays a significant role in kidney. We firstly tested the expression of PPP3CB in normal renal epithelial cells (HK2) and epithelial-like tumor cells (G401). The results showed that the expression of PPP3CB was the same level in G401 cells and HK2 cells (Figure 2A,B). EMT is a complex and multistep process, which occurs as a result of several molecular alterations. These molecular changes facilitate tumor cell migration from the primary site to distant sites [3,4]. Therefore, to explore the potential role of PPP3CB in the process of EMT, we overexpressed PPP3CB in G401 cells and evaluated the level of EMT markers. Overexpression of PPP3CB upregulated epithelial marker E-cadherin and downregulated mesenchymal markers *N*-cadherin, Vimentin, and Snail1, but had no effect on Twist1 (Figure 2C,D). Next, we knocked down PPP3CB in G401 cells. We found that the loss of PPP3CB generated more elongated cells compared with control (Figure 2E). To further confirm the change in cell morphology, we stained F-actin with phalloidin. The immunofluorescent staining clearly showed that the loss of PPP3CB caused actin cytoskeleton reorganization (Figure 2F). Additionally, this phenotypic switch was consistent with the significant reduction in mRNA and protein expression of epithelial marker E-cadherin and the significant increase of mesenchymal marker Vimentin (Figure 2G,H). Together, these results suggest that PPP3CB inhibits EMT in G401 cells.

### 2.3. PPP3CB Inhibits Migration of G401 Cells

EMT is correlated with tumor cell motility, invasion, and enhanced metastasis [16]. We next examined the effect of PPP3CB overexpression or knockdown on the migration of G401 cells. The wound healing scratch assays and migration Transwell assay showed that the overexpression of PPP3CB inhibited migration of G401 compared with the control group (Figure 3A,B). On the contrary, the loss of PPP3CB increased the wound closure rate and migration rate contributing to the migration of G401 cells (Figure 3C,D). Taken together, the results indicate that PPP3CB represses migration of G401 cells.

### 2.4. PPP3CB Promotes Cell Proliferation

We next explored whether PPP3CB is involved in tumor proliferation. We overexpressed PPP3CB in G401 cells. Cell proliferation was assessed by different methods. The results showed that overexpression of PPP3CB promoted cell growth (Figure 4A,B). Conversely, loss of PPP3CB inhibited G401 cell proliferation (Figure 4C,D). In vivo, 5 × 10^6^ G401 stable cells with sh-NC or sh1-PPP3CB were injected subcutaneously into athymic nude mice (six mice per group). Five out of six mice had a palpable tumor 7 weeks after inoculation with sh-NC cells. Tumor growth occurred only in four out of six mice injected with sh1-PPP3CB cells. We observed that the tumor volume of nude mice which were injected with G401 stable cells without PPP3CB was significantly smaller compared with the control (Figure 4E,F). We also investigated whether PPP3CB affected apoptosis of G401 cells. We overexpressed or knocked down PPP3CB in G401 cells and detected cell apoptosis with flow cytometry. The results showed that PPP3CB did not affect G401 cell apoptosis. (Figure A2). In this study, we found that overexpression of PPP3CB promoted G401 cell growth in vitro and in vivo. To further understand the function of PPP3CB, we used bioinformatic analysis. The analysis results of Kaplan-Meier Plotter and SurvExpress showed that high expression of PPP3CB was associated with worse survival in various tumors such as bladder carcinoma, ovarian cancer, gliomas, and glioblastoma (Figure A3). Similar to the with bioinformatic results, previous research found that high PPP3CB expression was an independent indicator predicting poor prognosis of neuroblastoma (NB) [17]. These results indicate that PPP3CB may be a crucial indicator in the diagnosis or prognosis of cancer.

### 2.5. PPP3CB 1-401 Containing the Catalytic Domain Plays a Critical Role in EMT and Cell Proliferation

PPP3CB is a serine/threonine phosphatase with four domains: a catalytic domain, a helical calcineurin B-binding segment, a calmodulin-binding segment, and an autoinhibitory peptide (Figure 5A). Once activated, PPP3CB combines with the target proteins and dephosphorylates substrate proteins, then promotes signal transduction [11]. For example, the catalytic domain of PPP3CB interacts with ATOH8 and dephosphorylates it, which influences the localization of ATOH8 [12]. Therefore, to investigate whether the catalytic domain of PPP3CB plays a vital role in EMT and cell proliferation, we deleted the PPP3CB 1–401 region, containing the phosphatase domain (Figure 5A). The results showed that deletion of PPP3CB 1–401 had no effect on the expression of E-cadherin (Figure 5B). In addition, the deletion of PPP3CB 1–401, remarkably disrupted PPP3CB-inhibited cell migration (Figure 5C). Also, we found that cell proliferation did not change after PPP3CB 402-524- overexpressing cells compared with the control (Figure 5D,E). Together, these findings indicate that PPP3CB 1–401 is an essential region for EMT of G401 cells and cell growth.

## 3. Discussion

The members of the PPP family have been extensively studied in EMT of cancer via regulating multiple signal pathways, such as TGF-β and WNT [8]. Recently, we found that PPP3CB may be a regulator of EMT. The underlying roles of PPP3CB in EMT have not been fully elucidated. In our study, we found that PPP3CB could suppress the EMT process and inhibit cell migration. In addition, PPP3CB facilitated tumor cell growth in vivo and in vitro. Furthermore, the phosphatase domain of PPP3CB was demonstrated to be the key region in regulating cell migration, the EMT process, and cell proliferation. These results indicate that PPP3CB might be an important regulator in the metastasis of cancer.

EMT is integral to the increased invasion and metastatic capabilities of tumor cells. It is well-known that EMT is typically characterized by the loss of the epithelial cell marker E-cadherin and the acquisition of mesenchymal markers Vimentin and *N*-cadherin [4,18]. During the process of EMT, epithelial cells reorganize their cytoskeleton and undergo an alteration in cell shape, which promotes the motility of individual cells and endows the development of an invasive phenotype [18]. Here, we originally demonstrated that knock-down of PPP3CB promoted EMT and rearrangement of F-actin (Figure 2), leading to accelerated cell migration (Figure 3). Our data suggest that inhibition of PPP3CB could significantly enhance invasion and migration via regulating EMT.

PPP3CB belongs to the catalytic subunit of calcineurin, which can dephosphorylate its substrates to modulate their physiological activities. For example, the catalytic domain of PPP3CB interacts with and dephosphorylates ATOH8, which results in translocation of ATOH8 from the cytoplasm to the nucleus [12]. To explore the underlying mechanisms of how PPP3CB regulates the EMT process and cell proliferation, we constructed a truncated PPP3CB mutant by deletion of the phosphatase domain (Figure 5). The results showed that loss of phosphatase catalytic activity had no effect on EMT, cell migration, or tumor cell proliferation, which indicates that the phosphatase domain of PPP3CB plays a crucial role in EMT and cell growth. Interestingly, although PPP3CB could obviously inhibit the migration of tumor cells and rearrange the cytoskeleton, we also found that PPP3CB promoted tumor cell growth. This is consistent with a previous study which showed that repression of EMT contributed to increased cancer cell proliferation in tumors [19].

PPP3CB serves as a key regulator of EMT in G401 cells, and the phosphatase domain contributes to the regulation of EMT and cell proliferation. We analyzed the protein interactions of PPP3CB by string software. The results showed that there were multiple molecules interacting with PPP3CB, including NFATc1, NFATc2, NFATc3, PPP1CA, and PPP2CA (Figure A4). In accordance with the string analysis results, studies show that PPP3CB interacts with various target substrates, such as NFATc1, NFATc2, and KSR2 [11]. PPP3CB dephosphorylates NFATc1 and NFATc2. Constitutively active NFATc2 induces apoptosis, whereas constitutively active NFATc1 promotes proliferation and transformation [20]. These findings suggest that PPP3CB may dephosphorylate substrates to regulate EMT and cell proliferation in tumor cells. Therefore, further investigations are needed to clarify the target substrate protein of PPP3CB involved in the EMT process and cell growth. If the key role of phosphatase activity and direct targeting of PPP3CB are confirmed, it would provide a novel approach (phosphatase inhibitor) and potential molecular target for therapy of metastatic tumors.

## 4. Materials and Methods

### 4.1. Plasmids Construction

The h. PPP3CB CDS was amplified from cDNA of G401 cells. The primer: forward, 5′-ggatccgaattcTGGCCGCCCCGGAG-3′ and reverse, 5′-atggtggtgctcgagTCACTGGGCAGTATGGTT GCC-3′ was used for PCR. The h. PPP3CB 402–524 was amplified from cDNA of G401 with the forward primer: 5′-ggatccgaattcGATGTAGGTTCAGCTGCAGCC-3′ and the reverse primer: 5′-atggtggtgctcgagTCACTGGGCAGTATGGTTGC-3′. The PCR products were cloned into the pCMV-GFP vector using EcoR*I* and Xho*I* sites. shRNA specific sequence targeting human PPP3CB were used (target1: CCTGCTAATACACGATACCTT, target2: GTCACAATACAGTTCGAGGAT). The PPP3CB shRNA oligos were synthesized by Sangon Biotech and inserted into the pLKO.1 vector at the site of EcoR*I* and Age*I*.

### 4.2. Cell Culture and Transfection

mK3, mK4, and 293FT cells were cultured in Dulbecco’s Modified Eagle Medium (DMEM) (Gibco, Carlsbad, CA, USA) supplemented with 10% fetal bovine serum (FBS) (Bioind, Kibbutz Beit Haemek, Israel) and 1% penicillin/streptomycin (PS) (Invitrogen, Grand Island, NY, USA). HK2 cells were cultured in Dulbecco’s Modified Eagle Medium: Nutrient Mixture F-12 (DMEM/F-12) (Gibco, Carlsbad, CA, USA) supplemented with 10% FBS and 1% PS. The G401 cells were maintained in Modified McCoy’s 5a Medium (Bioind, Kibbutz Beit Haemek, Israel) with 10% FBS and 1% PS. All the cell lines were cultured in a 37 °C, 5% CO_2_ incubator. Cells were transiently transfected using Linear Polyethylenimine according to the manufacturer’s instructions.

### 4.3. Construction of Stable Cell Lines

Lentivirus production was performed according to the manufacturer’s instructions. Briefly, lentiviral target plasmid (20 μg) with PSPA (15 μg) and PMD2G (5 μg) were transfected into 293FT cells to generated viruses. The viral particles were collected at 48 h post-transfection and filtered with a 0.45 μm filter. Viral particles were used to infect the G401 cells with polybrene. After transfection for 24 h, the stable cells were screened by puromycin (invivogen, San Diego, CA, USA).

### 4.4. RNA Extraction and Real-Time PCR

Total RNA was extracted from the cells with Trizol reagent (Invitrogen, Carlsbad, CA, USA). The cDNA was prepared by transcription kit (Thermo Scientific, Waltham, MA, USA) using oligonucleotide (dT) primers. The expression of target genes was detected by Ultra SYBR Mixture (CWBIO, Beijing, China). Data were normalized to internal control (18 s) for each experiment. QPCR primers were described as follows: hppp3cb: forward, 5′-CCCCAACACATCGCTTGACAT-3′ and reverse, 5′-GGCAGCACCCTCATTGATAATTC-3′ hcdh1: forward, 5′-CACCCTGGCTTTGACGCC GA-3′ and reverse, 5′-AAAATTCACTCTGCCCAGGACGCG-3′ hcdh2: forward, 5′-CGCCATCCA GACCGACCCAA-3′ and reverse, 5′-GTCGATTGGTTTGACCACGGTGAC-3′, hsnail1: forward, 5′-TCGGAAGCCTAACTACAGCGA-3′ and reverse, 5′-AGATGAGCATTGGCAGCGAG-3′ htwist1: forward, 5′-CACGAGCGGCTCAGCTACGC-3′ and reverse, 5′-AATGACATCTAGGTCTCCGGC CC-3, hvimentin: forward, 5′-GACGCCATCAACACCGAGTT-3, and reverse, 5′-CTTTGTCGTTGG TTAGCTGGT-3′. mcdh1: forward, 5′-CAGGTCTCCTCATGGCTTTGC-3′ and reverse, 5′-CTTC CGAAAAGAAGGCTGTCC-3′ mvimentin: forward, 5′-GGATCAGCTCACCAACGACA-3′ and reverse, 5′-AAGGTCAAGACGTGCCAGAG-3′ mtwist1: forward, 5′-GCCGGAGACCTAGATGTC ATT-3′ and reverse, 5′-CCACGCCCTGATTCTTGTGA-3′ msnail1: forward, 5′-AGCCCAACTATAG CGAGCTG-3′ and reverse, 5′-CCAGGAGAGAGTCCCAGATG-3′ mppp3cb: forward, 5′-AAAGCG TGCTGACACTCAAG-3′ and reverse, 5′-TGGAGAGAATCCTCGTATTGCT-3′.

### 4.5. Western Blotting

Proteins were extracted using 1% SDS lysis buffer. The concentration of proteins was measured with the Pierce BCA Protein Assay Kit (Thermo Scientific, Waltham, MA, USA) according to the manufacturer’s instructions. Each sample was separated by SDS-PAGE gel, transferred into nitrocellulose membrane (Millipore Corporation, Billerica, MA, USA). Membranes were incubated with the following primary antibodies: anti-PPP3CB (Abcam, Cambridge, UK), anti-E-cadherin (Bioworld, Nanjing, China), anti-Vimentin (Cell Signaling Technology, Danvers, MA, USA), anti-N-cadherin (Abcam, Cambridge, UK), anti-Twist1(Abcam, Cambridge, UK),anti-Snail1(Cell Signaling Technology, Danvers, MA, USA),anti-GFP (Bioworld, Nanjing, China), anti-β-actin (Transgene, Beijing, China). The target bands were detected by ECL (Millipore Corporation, Billerica, MA, USA).

### 4.6. Scratch Wound Healing Assay

The stable G401 cells were cultured in 6-well plates in Modified McCoy’s 5a Medium with 10% fetal bovine serum and 1% penicillin/streptomycin. When the confluence reached 90–100%, the monolayer of cells was scratched using a pipette tip to produce a wound. Then, 1 × PBS washed cells for three times. The cells were cultured in serum free Modified McCoy’s 5a Medium in a 37 °C, 5% CO_2_ incubator. The images were acquired at different time points (0 h, 12 h, 24 h) using a fluorescence microscope (ECLIPSE Ti-s, Nikon, Tokyo, Japan). images were collected from four independent field at each sample.

### 4.7. Migration Assay

3 × 10^4^ cells of control and overexpression or knocking down of PPP3CB stable cells in 200 μL serum free Modified McCoy’s 5a Medium were seeded into an 8-μm pore membrane Transwell chamber (Millipore Corporation, Billerica, MA, USA). After culturing for 24 h at 37 °C, 4% paraformaldehyde, cells were fixed on the lower surface for 20 min. Cells were washed with PBS three times. Cells were stained with 1% crystal violet for 20 min. Next, the cells were washed a further three times with PBS. The stained cells were photographed and three independent fields were counted.

### 4.8. MTT Assay

Cell proliferation was determined by MTT assay according to the manufacturer’s instructions. Briefly, 5 × 10^3^ stable G401 cells were cultured in 96 well plates in Modified McCoy’s 5a Medium with 10% fetal bovine serum and 1% penicillin/streptomycin. The cells were incubated in 90 μL serum free medium with 10 μL 5mg/mL MTT for 4 h. The medium was discarded. Dimethyl sulfoxide was used to dissolve the formazan crystals. The absorbance at 490 nm was measured.

### 4.9. 5-Ethynyl-21-Deoxyuridine (EdU) Assay

4 × 10^3^ stable G401 cells were cultured in 96 well plates in Modified McCoy’s 5a Medium with 10% fetal bovine serum and 1% penicillin/streptomycin. After 24 h, the proliferation of stable G401 cells was detected by the EdU DNA Proliferation in Detection kit (RiboBio, Guangzhou, China) according to the manufacturer’s instructions.

### 4.10. Flow Cytometry Assay

The stable G401 cells were cultured in 6 cm plates in Modified McCoy’s 5a Medium with 10% fetal bovine serum and 1% penicillin/streptomycin. When the confluence reached 90–100%, the cells were collected. Then, the stable G401 cell lines were stained with the Annexin V-fluorescein isothiocyanate (FITC) Apoptosis Detection Kit (KeyGEN BioTECH, Nanjing, China). Cell apoptosis was detected with flow cytometry. Flow cytometry was performed on 10,000 cells in each assay.

### 4.11. Nude Mice Tumorigenesis

5.0 × 10^6^ sh-NC and sh-PPP3CB-G401 cells in 200 μL PBS were injected subcutaneously into randomized athymic nude mice (six mice per group) [21]. Tumor growth was observed every day. Tumor volume was measured with calipers. Tumor volume (mm^3^) was calculated using the equation: *V* = length × width^2^/2. All mice were sacrificed at day 60. The volume of the excised tumors was measured.

### 4.12. Statistical Analysis

All experiments were performed independently at least three times and the results were presented as the mean ± standard error of the mean (SEM). The differences between two groups were analyzed using paired two-tailed Student’s *t*-test by GraphPad Prism 5 software (GraphPad Software, Inc., La Jolla, CA, USA). * *p* < 0.05, ** *p* < 0.01, and *** *p* < 0.001 indicate significant statistical differences compared with the control group.

## 5. Conclusions

The results shown in this study indicate that PPP3CB inhibits G401 cells migration through regulating EMT and promotes cell proliferation.

## Figures and Tables

**Figure 1 ijms-20-00275-f001:**
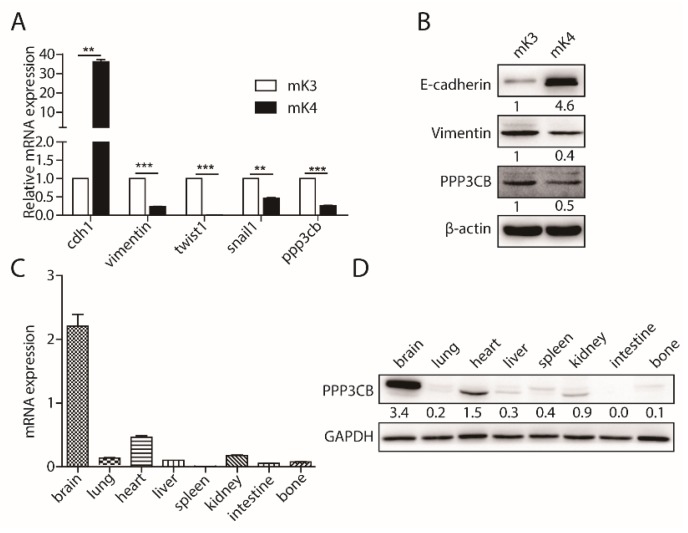
PPP3CB is an important regulator in kidney. (**A**) The indicated genes in mK3 and mK4 were detected by QPCR. Data represents the mean ± SEM of three independent experiments. * *p* < 0.05, ** *p* < 0.01, and *** *p* < 0.001. (**B**) The indicated proteins in mK3 and mK4 were detected by western blotting. (**C**,**D**) The expression of PPP3CB was tested in different tissues of mouse by QPCR and western blotting.

**Figure 2 ijms-20-00275-f002:**
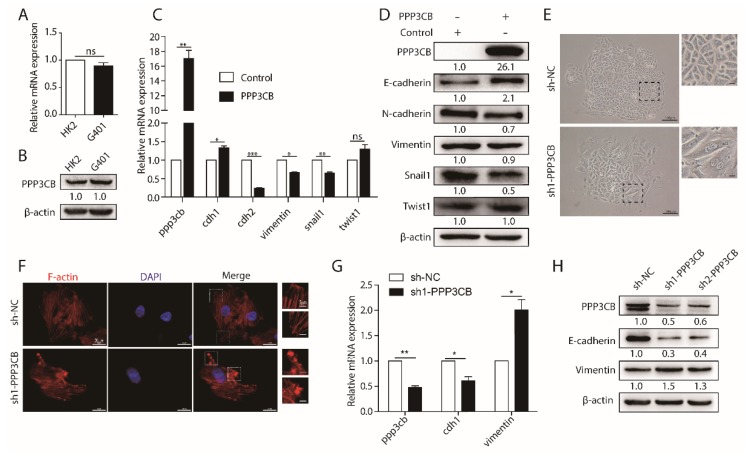
PPP3CB suppresses EMT of tumor cells. (**A**,**B**) Expression of PPP3CB in HK2 and G401 cells was detected by QPCR and western blotting. (**C**) Control or PPP3CB overexpressed G401 cells were subjected to QPCR with indicated genes. Data represents the mean ± SEM of three independent experiments. * *p* < 0.05, ** *p* < 0.01, and *** *p* < 0.001 (**D**) G401 cells infected with a lentivirus expressing Control and PPP3CB, subjected to western blotting with the indicated antibodies. (**E**) The images of G401 cells treated with sh-negative control (sh-NC) and sh1-PPP3CB, scar bar: 100 μm. The top-right subfigure of panel E means the magnified part, the scale bar is 35 μm. (F) Immunofluorescent staining of sh-NC and sh1-PPP3CB was assayed, red represents phalloidin, blue stains nucleus, scale bar is 20 μm. The top-right subfigure of panel F means the magnified part, the scale bar is 5 μm. (**G**) G401 cells with or without the depletion of PPP3CB, subjected to QPCR with indicated genes. Data represents the mean ± SEM of three independent experiments. * *p* < 0.05, ** *p* < 0.01. (**H**) G401 cells were treated with lentivirus vectors encoding two shRNA targeting PPP3CB or sh-NC, subjected to western blotting with indicated antibodies.

**Figure 3 ijms-20-00275-f003:**
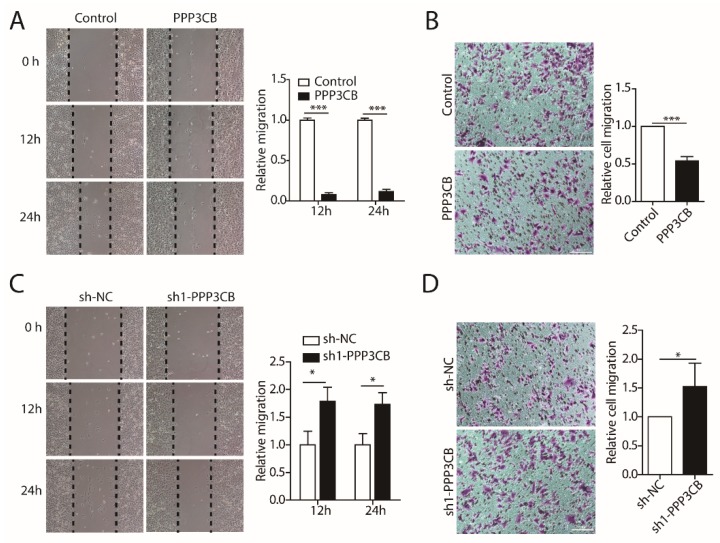
PPP3CB inhibits cell migration. (**A**,**C**) G401 stable cells with overexpression or knockdown of PPP3CB were assessed for cell migration by wound healing at the indicated time points (0 h, 12 h, and 24 h). Data were presented as mean ± SEM from three independent experiments. * *p* < 0.05, ** *p* < 0.01, and *** *p* < 0.001 (**B**,**D**) Transwell assays were used to assess cell migration. The scale bar is 100μm. Data were presented as mean ± SEM from three independent experiments. * *p* < 0.05, ** *p* < 0.01, and *** *p* < 0.001.

**Figure 4 ijms-20-00275-f004:**
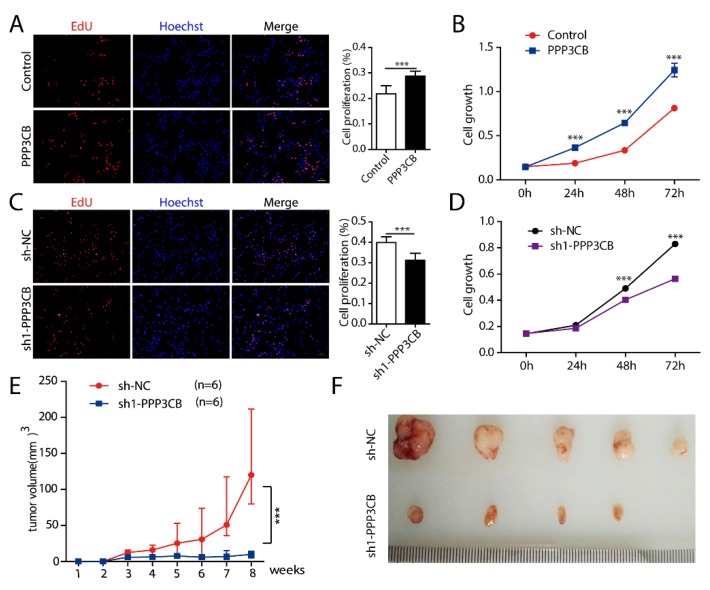
PPP3CB promotes G401 cell proliferation. (**A**,**C**) Growth of G401 stable cells was evaluated by EdU assay. Red represents EdU, blue stains nucleus, scale bar is 100 μm. Results were expressed as mean ± SEM from three replicate wells. * *p* < 0.05, ** *p* < 0.01, and *** *p* < 0.001. (**B**,**D**) Proliferation of G401 stable cells were assessed by MTT assay. Results were expressed as mean ± SEM from eight replicate wells. * *p* < 0.05, ** *p* < 0.01, and *** *p* < 0.001. (**E**) Xenograft tumor growth of G401 stable cells with sh-NC or sh1-PPP3CB in nude mice. Points indicate mean values (*n* = 6). * *p* < 0.05, ** *p* < 0.01, and *** *p* < 0.001. (**F**) Excised tumors.

**Figure 5 ijms-20-00275-f005:**
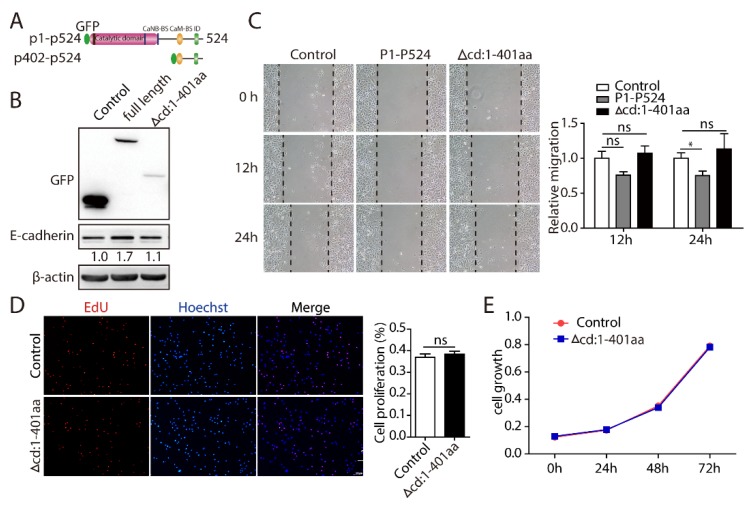
PPP3CB 1–401 containing the catalytic domain plays a critical role in EMT and cell proliferation. (**A**) PPP3CB structure and deletion (PPP3CB 402-524; Δcd:1–401 aa) (**B**) G401 cells infected with control, PPP3CB, and PPP3CB1–401 deletion, subjected to western blotting with indicated antibodies. (**C**) Cell migration of stable G401 cells with overexpression of PPP3CB or the PPP3CB1–401 deletion mutant assessed by wound healing at the indicated time points (0 h, 12 h, and 24 h). Data are presented as mean ± SEM from three independent experiments. * *p* < 0.05. (**D**) Growth of G401 stable cells were measured by EdU assay. Red represents EdU, blue stains nucleus, scale bar is 100 μm. Results are expressed as mean ± SEM from three replicate wells. (**E**) Proliferation of stable G401 cells were measured by MTT assay. Results were expressed as mean ± SEM from eight replicate wells.

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
