# Peer review of "PPP3CB Inhibits Migration of G401 Cells via Regulating Epithelial-to-Mesenchymal Transition and Promotes G401 Cells Growth"

_ijms, 2019, doi:10.3390/ijms20020275_

Round 1
Reviewer 1 Report
The Authors demonstrated that PPP3CB inhibits tumor cell migration through regulating EMT and promotes cell migration.
The paper is well organized and written. It can be acceptable after minor revisions.
The authors should discuss in more details how their study can be used in clinical stidies, if the evaluation of PPP3CB can improve diagnosis or prognosis of diseases and cancers and if there are information in literature and in clinical databases about its expression and the patient survival.
Author Response
Our response to reviewers' comments is attached.

Reviewer 2 Report
The re-submitted manuscript by Lei Chen et al. entitled "PPP3CB Inhibits Migration via Regulating Epithelial to-Mesenchymal Transition and Promotes Cells Growth", which I previously reviewed and rejected due to misinterpretation on G401 cell line properties, has been now properly corrected by authors. However, some major issues still need to be fixed.
- The authors claimed PPP3CB as "potent regulator of EMT" which has been screened among several "unknown regulators" of EMT by quantitative proteomic assay performed on mouse kidney epithelial (i.e. mK4) and mesenchymal (i.e. mK3) cell lines. Unfortunately, these quantitative proteomic data have not been showed by authors: it's pivotal to show these data in order to sustain the choice to further investigate the EMT role played by PPP3CB on G401 human rhabdoid tumor cell line.
- In the present study the authors explored the contribution of PPP3CB expression to the process of EMT assessing the mRNA and protein levels of E-cadherin and Vimentin on G401 cells. Their experimental approach was substantially correct: E-cadherin is regarded as a marker of epithelial cells, while Vimentin is responsible for stabilization of the cytoskeleton. However, the authors should also investigate in control and PPP3CB overexpressing G401 cells, the mRNA and protein levels of other molecules more directly involved in the EMT process: 1) N-cadherin, which is regarded as a marker of mesenchymal cells; 2) Snail and Twist, which are regulators of E-cadherin and N-cadherin expression.
- While the RT-PCR results are easily readable throughout the manuscript figures, the same can not be said about the western-blot plots. I strongly suggest to quantify the immunoblots showed in Figure 1B, Figure 2D, and Figure 2H using an image analysis method (e.g. Image J software), in order to make them promptly readable.
- Since almost all PPP3CB findings are referred to G401 cells, the name of this tumor cell line should be also mentioned in the paper title.
- The authors should also report the distribution of PPP3CB in human tissues citing data from one of the human-platforms available online (e.g. The Human Protein Atlas) in order to show the distribution of PPP3CB across all major tissues and organs in the human body.
Author Response
Our response to reviewer's comments is attached

Round 2
Reviewer 2 Report
In the revised version of this manuscript entitled "PPP3CB Inhibits Migration of G401 Cells via Regulating Epithelial-to-Mesenchymal Transition and Promotes G401 Cells Growth", by Lei Chen et al., the authors addressed all my past concerns raised during the previous stage of the revision, and they provided an improved manuscript that is now suitable for publication on IJMS.
This manuscript is a resubmission of an earlier submission. The following is a list of the peer review reports and author responses from that submission.
Round 1
Reviewer 1 Report
The Authors reported a study on the role of PPP3CB in EMT. The paper is interesting and evidences results by in vitro and in vivo experiments.
The Authors should revise the manuscript in some points before of publication:
- they must explain why they selected three cell lines, what are their properties.
- they should improve the paragraph of in vivo experiments explaining in more details their experiment (for example number of mice as reported in methods)
- they should insert some details on PPP3CB by inserting in the discussion an interactomic analysis (for example by string software) to underline if and with what proteins PPP3CB interacts and to hypothesize further studies on the evaluation of other proteins and the research of other markers. It could improve the conclusions of this paper
- they should revise the English. There are some mistakes in the text
Reviewer 2 Report
Comments:
1. Lines 75-76, why did the authors cite this reference (#14) here? It is not easy understood.
2. As shown the upper image in Fig.2C, the G401 (transfect sh-NC as control) showed the epithelial-like cell morphology that implied it does not tread to metastasis. Therefore, what is the basal level of PPP3BC in G401 cells? If possible, please compare PPP3BC level in G401 cells with normal renal epithelial cells.
3. Did deletion of PPP3BC 1-401 involve in regulating cell proliferation?
4. In Fig. 5E, there are 6 mice in each group, but why just only 5 tumors were showed in Fig. 5F?
5. The discussion section is poor.
Reviewer 3 Report
In this manuscript entitled "PPP3CB Inhibits Epithelial-to-Mesenchymal Transition but Promotes Tumor Growth in Wilms Tumor", by Lei Chen et al., the authors aimed to investigate the role of PPP3CB, one of the three isoforms of the catalytic A subunit of the calcium/calmodulin-dependent serine/threonine phosphatase (i.e. Calcineurin A), in the epithelial-mesenchymal transition (EMT) and tumor growth of Wilms tumor. They performed their experiments using two mouse kidney cell lines (i.e. MK3 and MK4) and one human cancer cell line (i.e. G401). They found that overexpression of PPP3CB is associated with the inhibition of the EMT and migration of G401 cells, and that the loss of PPP3CB is associated with suppression of the G401 cells growth both in vitro and in vivo experiments.
Unfortunately, this paper is not suitable for publication. The G401 cell line has been utilized by authors to demonstrate the role of PPP3CB in Wilms' tumor. However, the G401 cell line has properties of a rhabdoid phenotype of the kidney rather than that of a Wilms' tumor (Garvin AJ, et al. The G401 cell line, utilized for studies of chromosomal changes in Wilms' tumor, is derived from a rhabdoid tumor of the kidney. Am J Pathol. 1993 Feb;142(2):375-80). The rhabdoid tumors of the kidney is characterized by highly aggressive behavior and resistance to chemotherapy than Wilms' tumor and it's no longer included, as it was in 1970s, under the broad category of Wilms' tumors. Overall, the authors massively misinterpreted their results about the role PPP3CB in Wilms' tumor.
Round 2
Reviewer 2 Report
This manuscript can be publish in IJMS.
Reviewer 3 Report
The revised paper can not be accepted for the reasons reported after the first stage of review: the authors massively misinterpreted their results about the role PPP3CB in Wilms' tumor. They used the G401 cell line that has properties of a rhabdoid phenotype of the kidney rather than that of a Wilms' tumor... G401cell line served as a malignant rhabdoid tumor cell lines in plenty of studies...see refs reported below:
- Li Y, et al. Silibinin inhibits migration and invasion of the rhabdoid tumor G401 cell line via inactivation of the PI3K/Akt signaling pathway.
Oncol Lett. 2017 Dec;14(6):8035-8041.
- Unland R, et al. Analysis of the antiproliferative effects of 3-deazaneoplanocin A in combination with standard anticancer agents in rhabdoid tumor cell lines.
Anticancer Drugs. 2015 Mar;26(3):301-11.
- Algar EM, et al. Imprinted CDKN1C is a tumor suppressor in rhabdoid tumor and activated by restoration of SMARCB1 and histone deacetylase inhibitors.
PLoS One. 2009;4(2):e4482.
- Katsumi Y, et al. Sensitivity of malignant rhabdoid tumor cell lines to PD 0332991 is inversely correlated with p16 expression.
Biochem Biophys Res Commun. 2011 Sep 16;413(1):62-8.
- Krust B, et al. Suppression of tumorigenicity of rhabdoid tumor derived G401 cells by the multivalent HB-19 pseudopeptide that targets surface nucleolin.
Biochimie. 2011 Mar;93(3):426-33.
- Megison ML, et al. FAK inhibition abrogates the malignant phenotype in aggressive pediatric renal tumors.
Mol Cancer Res. 2014 Apr;12(4):514-26.
- Zhang K, et al. Frequent overexpression of HMGA2 in human atypical teratoid/rhabdoid tumor and its correlation with let-7a3/let-7b miRNA.
Clin Cancer Res. 2014 Mar 1;20(5):1179-89.
- Koos B, et al. The tyrosine kinase c-Abl promotes proliferation and is expressed in atypical teratoid and malignant rhabdoid tumors.
Cancer. 2010 Nov 1;116(21):5075-81.
- Lünenbürger H, et al. Systematic analysis of the antiproliferative effects of novel and standard anticancer agents in rhabdoid tumor cell lines.
Anticancer Drugs. 2010 Jun;21(5):514-22.
- Yanagisawa S, et al. Identification and metastatic potential of tumor-initiating cells in malignant rhabdoid tumor of the kidney.
Clin Cancer Res. 2009 May 1;15(9):3014-22.